# The retinoid X receptor has a critical role in synthetic rexinoid-induced increase in cellular all-*trans*-retinoic acid

Olga V. Belyaeva[1], Alla V. Klyuyeva[1], Ansh Vyas[1], Wilhelm K. Berger[2], Laszlo Halasz[2], Jianshi Yu[3], Venkatram R. Atigadda[4], Aja Slay[1], Kelli R. Goggans[1], Matthew B. Renfrow[1], Maureen A. Kane[3], Laszlo Nagy[2], Natalia Y. Kedishvili[1]*

1 Department of Biochemistry and Molecular Genetics, Heersink School of Medicine, University of Alabama at Birmingham, Birmingham, AL, United States of America, 2 Departments of Medicine and Biological Chemistry, Johns Hopkins University School of Medicine, Institute for Fundamental Biomedical Research, Johns Hopkins All Children's Hospital, St. Petersburg, FL, United States of America, 3 Department of Pharmaceutical Sciences, School of Pharmacy, University of Maryland, Baltimore, MD, United States of America, 4 Department of Dermatology, Heersink School of Medicine, University of Alabama at Birmingham, Birmingham, AL, United States of America

* nkedishvili@uab.edu

**Data Availability Statement:** Sequencing data sets performed in this study are available at the NCBI GEO under accession numbers: GSE244387. GEO

## Abstract

Rexinoids are agonists of nuclear rexinoid X receptors (RXR) that heterodimerize with other nuclear receptors to regulate gene transcription. A number of selective RXR agonists have been developed for clinical use but their application has been hampered by the unwanted side effects associated with the use of rexinoids and a limited understanding of their mechanisms of action across different cell types. Our previous studies showed that treatment of organotypic human epidermis with the low toxicity UAB30 and UAB110 rexinoids resulted in increased steady-state levels of all-*trans*-retinoic acid (ATRA), the obligatory ligand of the RXR-RAR heterodimers. Here, we investigated the molecular mechanism underlying the increase in ATRA levels using a dominant negative RXRα that lacks the activation function 2 (AF-2) domain. The results demonstrated that overexpression of dnRXRα in human organotypic epidermis markedly reduced signaling by resident ATRA, suggesting the existence of endogenous RXR ligand, diminished the biological effects of UAB30 and UAB110 on epidermis morphology and gene expression, and nearly abolished the rexinoid-induced increase in ATRA levels. Global transcriptome analysis of dnRXRα-rafts in comparison to empty vector-transduced rafts showed that over 95% of the differentially expressed genes in rexinoid-treated rafts constitute direct or indirect ATRA-regulated genes. Thus, the biological effects of UAB30 and UAB110 are mediated through the AF-2 domain of RXRα with minimal side effects in human epidermis. As ATRA levels are known to be reduced in certain epithelial pathologies, treatment with UAB30 and UAB110 may represent a promising therapy for normalizing the endogenous ATRA concentration and signaling in epithelial tissues.

token 'elofuqsqdvkvdwz' is generated for reviewer access. All other relevant data are within the manuscript and its Supporting Information files.

**Funding:** V. R. A., M. B. R., L. N. and N. Y. K. were supported by the UAB O'Neil Comprehensive Cancer Center https://nam12.safelinks.protection.outlook.com/?url=https%3A%2F%2Fwww.onealcanceruab.org%2F&data=05%7C02%7Cnkedishvili%40uab.edu%7C756526453 2cf44be85ad08dc47e49c33%7Cd8999fe476af40b3b4351d8977abc08c%7C1%7C1%7C638464294717398269%7CUnknown%7CTWFpbGZsb3d8eyJWIjoiMC4wLjAwMDAiLCJQIjoiV2luMzIiLCJBTiI6Ik1haWwiLCJXVCI6Mn0%3D%7C0%7C%7C%7C&sdata=yKvh5%2FAvpKNq%2BxcoX5NZYdEyLEY7mbCCCiI9RNwcwBg%3D&reserved=0 and the National Cancer Institute https://nam12.safelinks.protection.outlook.com/?url=https%3A%2F%2Fwww.cancer.gov%2F&data=05%7C02%7Cnkedishvili%40uab.edu%7C7565264532cf44be85ad08dc47e49c33%7Cd8999fe476af40b3b4351d8977abc08c%7C1%7C1%7C638464294717403363%7CUnknown%7CTWFpbGZsb3d8eyJWIjoiMC4wLjAwMDAiLCJQIjoiV2luMzIiLCJBTiI6Ik1haWwiLCJXVCI6Mn0%3D%7C0%7C%7C%7C&sdata=Mbq6srhwOr%2BbRgxFDZAbfvGOQF5MIxZ2Ps039OM1gic%3D&reserved=0 grant CA210946. L. N. was also supported by the National Institute of Diabetes and Digestive and Kidney Diseases https://nam12.safelinks.protection.outlook.com/?url=https%3A%2F%2Fwww.niddk.nih.gov%2F&data=05%7C02%7Cnkedishvili%40uab.edu%7C7565264532cf44be85ad08dc47e49c33%7Cd8999fe476af40b3b4351d8977abc08c%7C1%7C1%7C638464294717410038%7CUnknown%7CTWFpbGZsb3d8eyJWIjoiMC4wLjAwMDAiLCJQIjoiV2luMzIiLCJBTiI6Ik1haWwiLCJXVCI6Mn0%3D%7C0%7C%7C%7C&sdata=Mbq6srhwOr%2BbRgxFDZAbfvGOQF5MIxZ2Ps039OM1gic%3D&reserved=0 grant DK115924; M. A. K. was supported by the National Institute of Arthritis and Musculoskeletal and Skin Diseases https://nam12.safelinks.protection.outlook.com/?url=https%3A%2F%2Fwww.niams.nih.gov%2F&data=05%7C02%7Cnkedishvili%40uab.edu%7C7565264532cf44be85ad08dc47e49c33%7Cd8999fe476af40b3b4351d8977abc08c%7C1%7C1%7C638464294717414967%7CUnknown%7CTWFpbGZsb3d8eyJWIjoiMC4wLjAwMDAiLCJQIjoiV2luMzIiLCJBTiI6Ik1haWwiLCJXVCI6Mn0%3D%7C0%7C%7C%7C&sdata=UlWqQFTEu54UEHrIaosI0zudsdkfljHocEAucJyG6wk%3D&reserved=0 grant R01AR074846; University of Maryland, School of Pharmacy, P23474; and University of

## Introduction

The term 'rexinoids' encompasses all naturally occurring or synthetic compounds that serve as activating ligands for nuclear transcription factors retinoid X receptors (RXRα-γ (NR2B1-3)) [1–3]. While RXRs may function as homodimers or homotetramers [4, 5], RXRs can also heterodimerize with other nuclear receptors, including retinoic acid receptors (RARα-γ (NR1B1-3)), vitamin D receptor (VDR (NR1I1)), thyroid hormone receptor (TR (NR1A1-2)), liver X receptor (LXR (NR1H1-2)), and peroxisome proliferator-activated receptor (PPAR (NR1C1-3)) [1–3]. As heterodimeric partners with different nuclear receptors, RXRs can regulate numerous signaling pathways, playing essential and diverse roles in cell physiology. These properties of RXRs have been exploited for modulating cell metabolism, differentiation, proliferation, and survival. Synthetic rexinoids were originally developed for the treatment of metabolic disorders, but were subsequently explored as pharmacotherapy for cancers, Alzheimer's disease, Parkinson's disease, diabetes, inflammatory bowel disease, and autoimmune diseases [reviewed in 1]. Thus far, Targretin (bexarotene) is the only FDA approved rexinoid, and it is currently used to treat refractory cutaneous T-cell lymphoma [6]. However, Targretin retains some binding to RAR [7] and is known to induce hyperlipidemia in human patients when administered orally [reviewed in 8]. Other synthetic rexinoids have been developed that are more selective for RXR, with EC50s for RXR activation at nanomolar concentrations [reviewed in 2] including a series of UAB rexinoids developed by our group.

UAB30 is an RXR selective agonist that is currently being evaluated in human Phase I clinical trials by the National Cancer Institute [9]. Due to high efficacy and low toxicity UAB30 is being considered as preventative chemotherapy in women at high risk of developing breast cancer. To improve upon the potency of UAB30, a series of UAB30 homologues were generated with a single methyl group added to the carbon positions of the tetralone ring as well as inclusion of a disubstituted cyclohexenyl ring. Among the new compounds, UAB110 showed promise as it did not increase triglyceride levels in rat livers when dosed orally while being effective in the *in vivo* chemoprevention assay [10].

Although rexinoids have a high potential for pharmaceutical applications, their use has been limited due to poor understanding of the molecular events underlying their actions in different types of cells and tissues. For example, in human breast cancer cells, UAB30 was found to decrease DNA methyltransferase and telomerase expression [11]. Another study suggested that UAB30 and Targretin each inhibit invasion and migration by targeting Src in human breast cancer cells [12]. In neuroblastoma cell lines, potential activation of p53 by UAB30 might have been responsible for the UAB30-induced cellular differentiation [13]. As shown by studies from this laboratory, in a model of human organotypic epidermis, UAB30 selectively modulated the expression of genes regulated by RXR-RAR heterodimers while other RXR-mediated pathways were unaffected [14]. Interestingly, the enhanced signaling through RXR-RAR heterodimers induced by UAB30 as well as UAB110 was accompanied by an increase in the steady-state levels of all-*trans*-retinoic acid (ATRA) [14, 15], the obligatory ligand for RXR-RAR transcriptional activity. It is well-established that in RXR-RAR heterodimers, the binding of rexinoid to RXR can enhance the transcriptional activity of RXR-RAR heterodimers, but only if RAR is liganded with ATRA [16]. Hence, exogenously added rexinoids can potentiate the transcriptional activity of pre-existing resident ATRA in keratinocytes and, in addition, raise the levels of ATRA, further enhancing signaling through the RXR-RAR pathway.

The molecular mechanism underlying the increase in the steady-state levels of ATRA upon supplementation with RXR agonists UAB30 and UAB110 is unclear. Considering that rexinoids are structurally similar to endogenous retinoids and other polyisoprenoid compounds,

Maryland, School of Pharmacy Mass Spectrometry Center, SOP1841-IQB2014. N. Y. K. was also supported by the National Institute of Arthritis and Musculoskeletal and Skin Diseases https://nam12. safelinks.protection.outlook.com/?url=https%3A%2F%2Fwww.niams.nih.gov%2F&data=05%7C02%7Cnkedishvili%40uab.edu%7C7565264532cf44be85ad08dc47e49c33%7Cd8999fe476af40b3b4351d8977abc08c%7C1%7C1%7C638464294717419800%7CUnknown%7CTWFpbGZsb3d8eyJWIjoiMC4wLjAwMDAiLCJQIjoiV2luMzIiLCJBTiI6Ik1haWwiLCJXVCI6Mn0%3D%7C0%7C%7C%7C&sdata=WTjj8gQFFpqfzQ7yCyiJmG3KW4VTCVJmRW73idjvZ%2Bl%3D&reserved=0 grant R01AR076924. The funders, i.e., National Institutes of Health - National Cancer Institute; National Institute of Diabetes and Digestive and Kidney Diseases; and National Institute of Arthritis and Musculoskeletal and Skin Diseases - had no role in study design, data collection and analysis, decision to publish, or preparation of the manuscript.

**Competing interests:** The authors have declared that no competing interests exist.

the increase in ATRA could potentially result from competitive inhibition by rexinoids of either ATRA metabolizing enzymes, retinol esterifying enzymes, which sequester retinol in the storage form of retinyl esters, or because of interference with retinoic acid binding protein type 1 that channels ATRA to degradation by cytochrome P450 enzymes [17, 18]. To distinguish between the transcriptional *versus* metabolic effects of rexinoids, we employed a dominant negative variant of RXRα to reduce the transcriptional activity of endogenous native RXRα by removing the activation function (AF-2) domain. The results of this study clearly identify RXRα as the necessary element in mediating the increase in ATRA concentration and the resulting morphological and transcriptome changes in human organotypic epidermis.

## Materials and methods

### Preparation of constructs and skin raft cultures

The dominant negative form of murine RXRα lacking the C-terminal amino acids 449 through 467 and the transactivation function AF-2 domain was first described in [19]. Considering the nearly identical protein sequences of human and mouse RXRα, the same approach was applied to generate dominant negative human RXRα (dnRXRα). The cDNA encoding truncated human RXRα was amplified from the organotypic epidermis cDNA using primers 5'-AAA GAATTCATGGACACCAAACATTTCCTGCC-3' and 5'-AAACTCGAGCTACCCGATGAGCTTG AAGAAGAAG-3' (restriction sites underlined) and cloned into *EcoRI–SalI* sites of the pBabe-puro plasmid. The resulting construct lacks sequence encoding amino acids 444–462 (Fig 1A), which corresponds to the 449–467 amino acid segment of murine RXRα. The production of retroviral particles capable of infecting human primary keratinocytes (PHKs) was performed as described earlier [20].

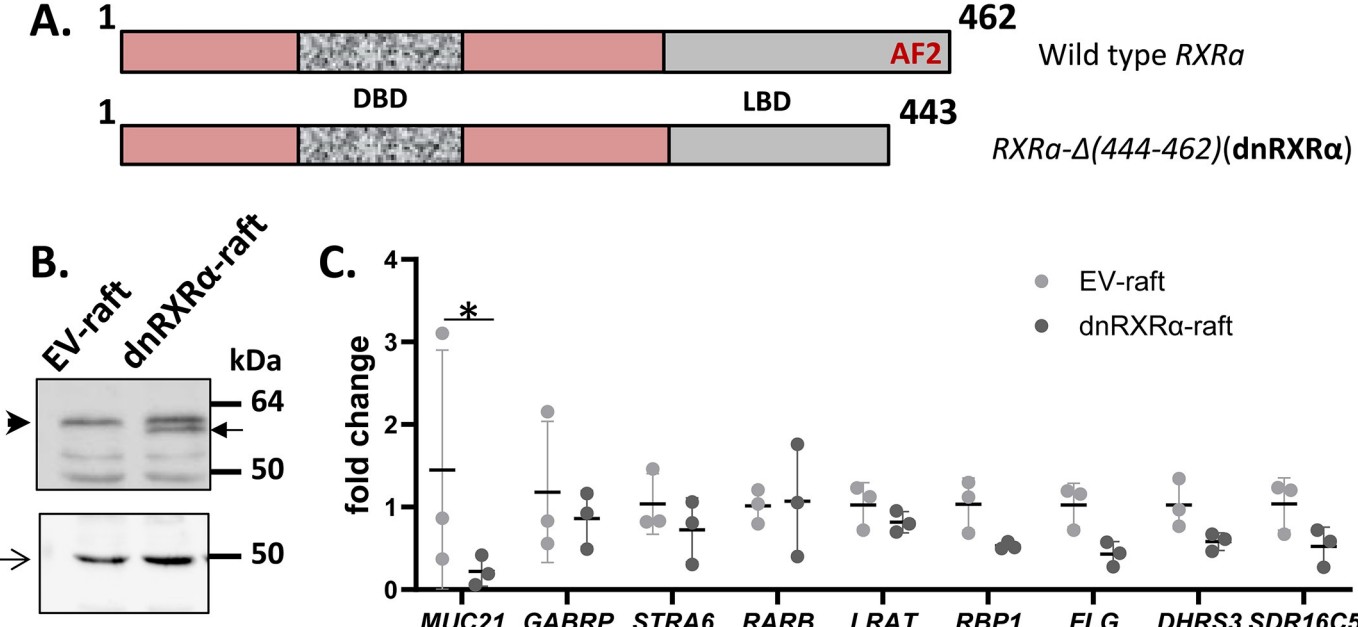

**Fig 1. Overexpression of dnRXRα in skin rafts and its effect on gene expression. A.** Schematic representation of the functional domains of dnRXRα protein compared to wild-type RXRα (not to scale). **B.** Western blot analysis of RXRα. Thirty μg of total extracts prepared from EV-rafts and dnRXRα-rafts were separated by electrophoresis in 10% polyacrylamide gel and immunostained with RXRα antibodies and β-actin antibodies (arrow) for loading control. The full-length RXRα is indicated by an arrowhead and dnRXRα by black arrow. Other visible bands resulted from non-specific binding of antibodies. **C.** QPCR analysis of gene expression in dnRXRα-rafts relative to EV-rafts; *p<0.05.

Neonatal foreskins for isolation of PHKs were obtained from the Newborn Nursery of the University of Alabama at Birmingham Hospital in compliance with University of Alabama at Birmingham Institutional Review Board (IRB) regulations. As determined by the institutional IRB, the use of discarded unidentifiable foreskin tissue met the requirements for an exemption from IRB approval. The PHKs were isolated from neonatal foreskins, cultured and infected with retrovirus carrying expression construct for dnRXRα (dnRXRα-raft) or an empty vector (EV) construct encoding only a puromycin resistance marker (EV-raft) as described before [20]. Following selection of transduced PHKs for the puromycin resistance encoded by the expression constructs, PHKs were differentiated into organotypic epidermis cultured at the liquid-air interface ("raft" cultures) [20]. Twelve hours before harvesting, BrdU was added to the culture medium from 100x stock to the final concentration of 50 μg/ml. Stratified cultures were harvested for analyses after 10 days of incubation at the liquid-air interface. For histological analysis, the whole rafts were fixed in 10% buffered formalin for 45 minutes, rinsed in 70% ethanol, dehydrated and embedded in paraffin; for isolation of RNA and analysis of retinoid content, epidermis was peeled off the collagen beds and frozen on dry ice.

## Treatment with rexinoids

From the day the skin equivalents were lifted onto the grids and until harvest, the raft culture medium was supplemented with either 2 μM UAB30, 0.2 μM UAB110 or vehicle (DMSO) added from 1000x stock. Supplementation with rexinoids was performed under dim light, and the medium was replaced every other day.

## Histology

H&E staining of skin rafts was performed essentially as described previously [20]. Briefly, paraffin-embedded skin rafts were cut into 5-μm sections, mounted on Superfrost/Plus slides (Fisher Scientific, Pittsburgh, PA), deparaffinized, rehydrated in a series of ethanol dilutions and stained with Harris hematoxylin (Fisher Scientific) and Eosin Y (Protocol, Fisher Scientific). Sections were mounted in Permount medium (Fisher Scientific) and analyzed at 10x and 20x magnification using AxioImager A2 microscope equipped with an AxioCam camera and AxioVision image capture software (Carl Zeiss MicroImaging, Inc., Thornwood, NY).

## Immunohistochemistry

For staining of BrdU-labeled nuclei, deparaffinized and rehydrated 5-μM sections were incubated in 3% hydrogen peroxide for 10 minutes to quench peroxidase activity, blocked in 2% bovine serum albumin in PBS with 0.1% Tween-20 for one hour, and incubated with 1:200 dilution of BrdU (ZBU30) antibody (03–3900, Invitrogen) at 4 °C overnight. Slides were washed in Tris-buffered saline with 0.1% Tween-20 (TBST) and incubated with a 1:50 dilution of MultiLink biotinylated secondary antibodies (SuperSensitive LinkLabel-IHC Detection Kit, Biogenex, San Ramon, CA) for 30 min at room temperature. Next, slides were washed with TBST and incubated with 1:50 dilution of streptavidine-conjugated horseradish peroxidase label followed by washes and incubation with ImmPACT NovaRed Peroxidase Substrate (Vector Laboratories). For staining against filaggrin, an antigen retrieval step was included after rehydration of the sections. Sections were submerged in 10 mM sodium citrate, pH 6.0, pre-heated to 65 °C, incubated overnight and processed as above. Mouse monoclonal antibody against filaggrin (NCL-FILAGGRIN, Leica Microsytems Inc. Bannockburn, IL) was used at a 1:200 dilution.

### Analysis of gene expression

RNA was isolated from individual rafts using Trizol reagent (Life Technologies) and transcribed using SuperScript III first strand synthesis system (Invitrogen). Real-time PCR analysis was conducted on Roche LightCycler®480 instrument (Roche Diagnostics) using LightCycler®480 SYBR Green I Master Mix (Roche, Indianapolis, IN). Relative gene expression levels were analyzed and normalized using geometric mean of expression levels of the reference genes as described before [20]. Three to four individual rafts were included in each qPCR experiment. To evaluate the significance of changes in the expression levels of each transcript between EV-rafts and dnRXRα-rafts, or DMSO and rexinoid-treated rafts, an unpaired *t* test was performed, and the two-tailed *p* value was determined using GraphPad Prism 8.

### Analysis of endogenous retinoid content

Retinoids were extracted from frozen skin raft samples under yellow light and quantified as described previously in detail [15]. Briefly, one skin raft per replicate was assayed. For internal standards, 2.2 μM all-*trans*-4,4-dimethyl-RA and 0.9 μM retinyl acetate were added to each sample. Retinol and retinyl esters per tissue weight were quantified by HPLC-UV using a Waters H-Class UPLC system equipped with a photodiode array detector operated in single wavelength detection mode according to previously published methodology [21, 22] ATRA concentration per tissue weight was determined by liquid chromatography-multistage-tandem mass spectrometry using a Shimadzu Prominence UFLC XR liquid chromatography system (Shimadzu, Columbia, MD) coupled to an AB Sciex 6500 QTRAP hybrid triple quadrupole mass spectrometer (AB Sciex, Framingham, MA) [23, 24].

### RNA-seq library construction and gene expression quantification

RNA-seq analysis was carried out as described previously in detail [15]. Briefly, RNA-Seq libraries were prepared from total RNA using Ultra II RNA Sample Prep kit (New England BioLabs) according to the manufacturer's protocol. Sequencing runs were executed on Illumina NextSeq 500 instrument using single-end 75 cycles sequencing. Sequencing quality of the single-ended mRNA reads was evaluated by *FastQC* software and were aligned to the human reference genome (*hg19*) using *HISAT2* [25] with default parameters. Genes were quantified using *featureCounts* [26]. Genes with at least 5 CPM mapped read were considered expressed. Statistically significant difference was considered with FDR < 0.05 and fold change > 1.5 using *edgeR* [27] package in R. Visualization of the results were performed in R.

## Results

### Overexpression of dnRXRα in organotypic skin raft culture

Epidermal keratinocytes express RXR α and β and RAR α and γ, but RXRα and RARγ are the predominant species present in all epidermal cell layers [reviewed in ref. 28]. Therefore, to evaluate the role of RXR in mediating the increase in ATRA, we employed a dominant negative form of RXRα (dnRXRα) to interfere with endogenous RXR activity in organotypic epidermis. DnRXRα lacks the activation function 2 (AF-2) domain but retains ligand-binding as well as dimerization functions of RXRα (Fig 1A) [29, 30].

To confirm that dnRXRα was successfully expressed in skin rafts, we performed western blot analysis of extracts prepared from EV-rafts (transduced with empty pBabe vector) and dnRXRα-rafts (transduced with dnRXRα-pBabe expression vector). Immunostaining using antibodies against RXRα showed an additional band in dnRXRα-rafts (Fig 1B) that ran slightly faster than the full-length RXRα, in agreement with the ~2 kDa lower molecular mass of dnRXRα.

To determine whether overexpression of dnRXRα altered the gene expression pattern in skin rafts, we performed qPCR analysis of the genes known to be regulated by ATRA. Out of nine genes tested, only one, *MUC21*, was found to be somewhat downregulated in dnRXRα-expressing rafts (Fig 1C). Considering our previous finding that *MUC21* is upregulated by ATRA [14], the decrease in *MUC21* expression suggested a possible attenuation of resident ATRA signaling in dnRXRα-rafts.

The impact of dnRXRα overexpression on skin rafts' histology was investigated using markers of keratinocytes proliferation and differentiation. To detect changes in proliferating basal keratinocytes, raft cultures were treated with BrdU for 24 hours before harvesting and skin raft sections were incubated with BrdU antibody to visualize its incorporation in proliferating nuclei. The differentiation of keratinocytes was assessed based on immunostaining for filaggrin, the marker of differentiated granular layer of epidermis. Overall, no significant changes were observed in the morphology and stratification of skin rafts overexpressing dnRXRα (Fig 2, dnRXRα +DMSO *versus* EV+DMSO), consistent with a relatively mild decrease in ATRA signaling as suggested by the results of qPCR analysis.

**Treatment with UAB30.** Having established that overexpression of dnRXRα did not visibly alter epidermal histology and had a mild impact on gene expression pattern, we next asked whether the overexpression of dnRXRα would impact the response of epidermal cultures to rexinoids. Our previous studies demonstrated that both UAB30 and UAB110 markedly altered the stratification pattern of skin raft cultures [14, 15]. In agreement with our previous results, treatment of skin rafts with 2 μM UAB30 increased the number of proliferating BrdU-positive basal keratinocytes and reduced the thickness of granular and cornified layers as indicated by the thinner and discontinuous pattern of immunostaining for filaggrin (Fig 2, EV+UAB30).

On the other hand, in dnRXRα-rafts, UAB30 appeared to have little or no effect on expression of filaggrin (Fig 2, dnRXRα+UAB30), and the number of BrdU-positive nuclei was noticeably lower than in UAB30-treated EV- rafts (Fig 2, EV+UAB30). However, even in dnRXRα-rafts, UAB30 treatment caused a higher BrdU incorporation than in DMSO-treated dnRXRα-rafts (Fig 2, compare dnRXRα+UAB30 with dnRXRα+DMSO), suggesting that dnRXRα was more effective in reducing the RXR signaling in the differentiated layers than in proliferating layers of the epidermal cultures.

**Treatment with UAB110.** UAB110 was previously found to be a much more potent rexinoid than UAB30 [10], and at 0.2 μM UAB110 induced changes in stratification of skin rafts similar to those induced by 10 μM UAB30 [15]. Indeed, as reproduced in the present study, treatment of EV-rafts with 0.2 μM UAB110 resulted in more profound changes in the morphology than treatment with a ten-fold higher concentration of UAB30 (2 μM). UAB110-treated rafts displayed vanishingly thin granular and cornified layers, nearly absent filaggrin staining, expanded spinous layer assuming sponge-like morphology, and a notable increase in BrdU-labeled nuclei (Fig 2, EV +UAB110). The density and the strength of desmosomal contacts between suprabasal cells were visibly reduced.

Remarkably, the presence of dnRXRα in skin rafts had largely prevented the strong impact of UAB110 on skin morphology (Fig 2, dnRXRα+UAB110). Staining for filaggrin demonstrated the presence of differentiated granular layer albeit somewhat thinner than in DMSO-treated rafts. The number of BrdU-positive nuclei was also significantly lower, suggesting the effect of UAB110 on basal cell proliferation was also reduced.

Together, these results provided strong evidence that a functional AF-2 domain of RXRα was necessary for mediating the effects of UAB30 and UAB110 on the proliferation and differentiation of PHKs and stratification of organotypic epidermis.

**Targeted analysis of gene expression in dnRXRα–rafts in the presence and absence of rexinoids.** To determine whether the reduced morphological response of dnRXRα-rafts to

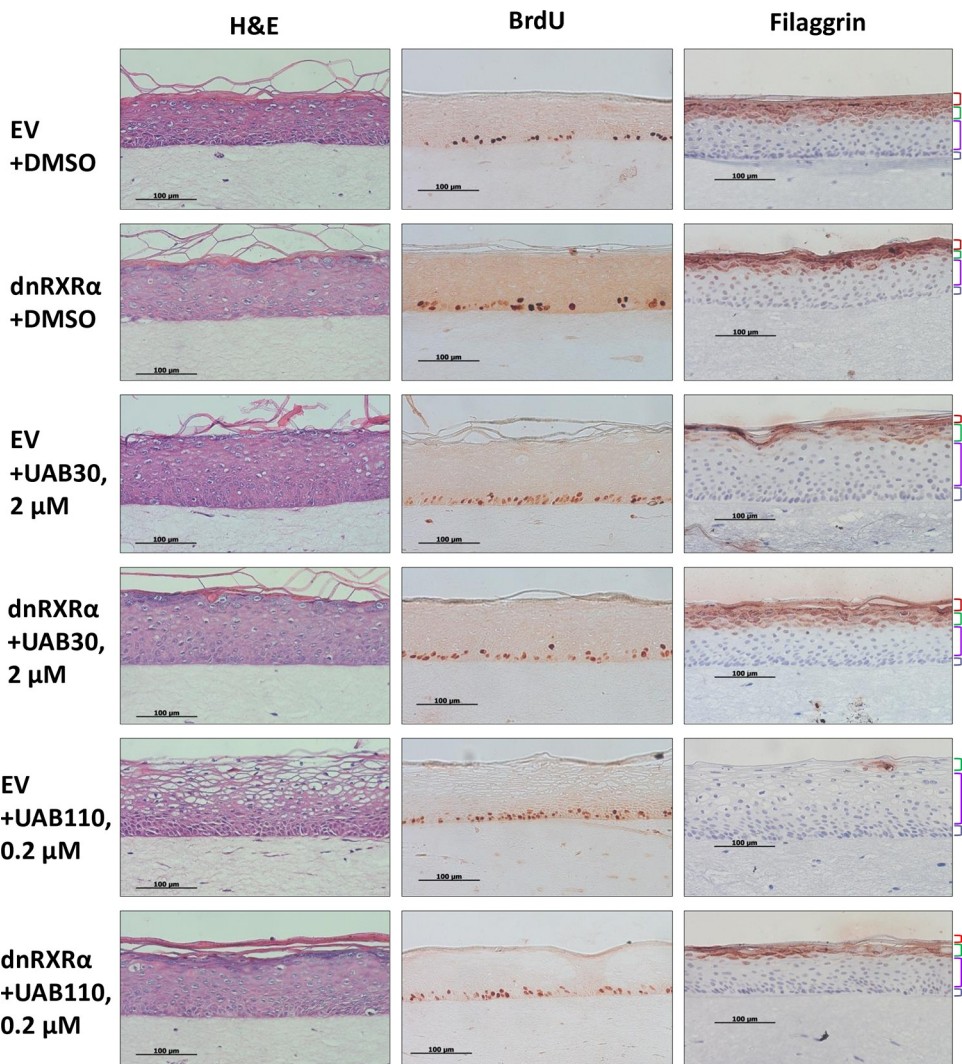

**Fig 2. Impact of dnRXRα on morphology of skin rafts treated with UAB rexinoids.** Skin rafts embedded in paraffin were sectioned and stained with H&E analysis of skin morphology, with BrdU antibodies for analysis of cell proliferation, and with filaggrin antibodies for analysis of cell differentiation as described under *Materials and Methods*. EV-rafts and dnRXRα-rafts were treated with either DMSO as a vehicle, with 2 μM UAB30, or with 0.2 μM UAB110. Differently colored brackets demarcate the layers of epidermis: cornified (red), granular (green), spinous (purple), and basal (gray). Note the increased thickness of the basal layer (gray bracket) and reduced differentiation to cornified layer (red bracket) in rafts treated with 2 μM UAB30 or 0.2 μM UAB110 as compared with DMSO-treated rafts.

rexinoids was due to reduced potency of ATRA signaling, we compared the expression of ATRA target genes in dnRXRα-rafts *versus* EV-rafts by qPCR. As expected, in EV-rafts, treatment with UAB30 caused a strong increase in expression of *GABRP* (23-fold), *MUC21* (32-fold), and *STRA6* (8-fold) transcripts; *RARB*, *LRAT* and *DHRS3* transcripts also showed significant upregulation (3–5 –fold) relative to DMSO-treated rafts (Fig 3, EV plus UAB30 and EV plus UAB110). Conversely, expression of genes negatively regulated by ATRA [14, 15, 20] was reduced: by 4-fold for *FLG* and by 2.6-fold for *SDR16C5*.

In comparison, UAB30 treatment had a much weaker effect on ATRA-responsive transcripts in dnRXRα-rafts (Fig 3, dnRXRα plus UAB30 and dnRXRα plus UAB110). Expression

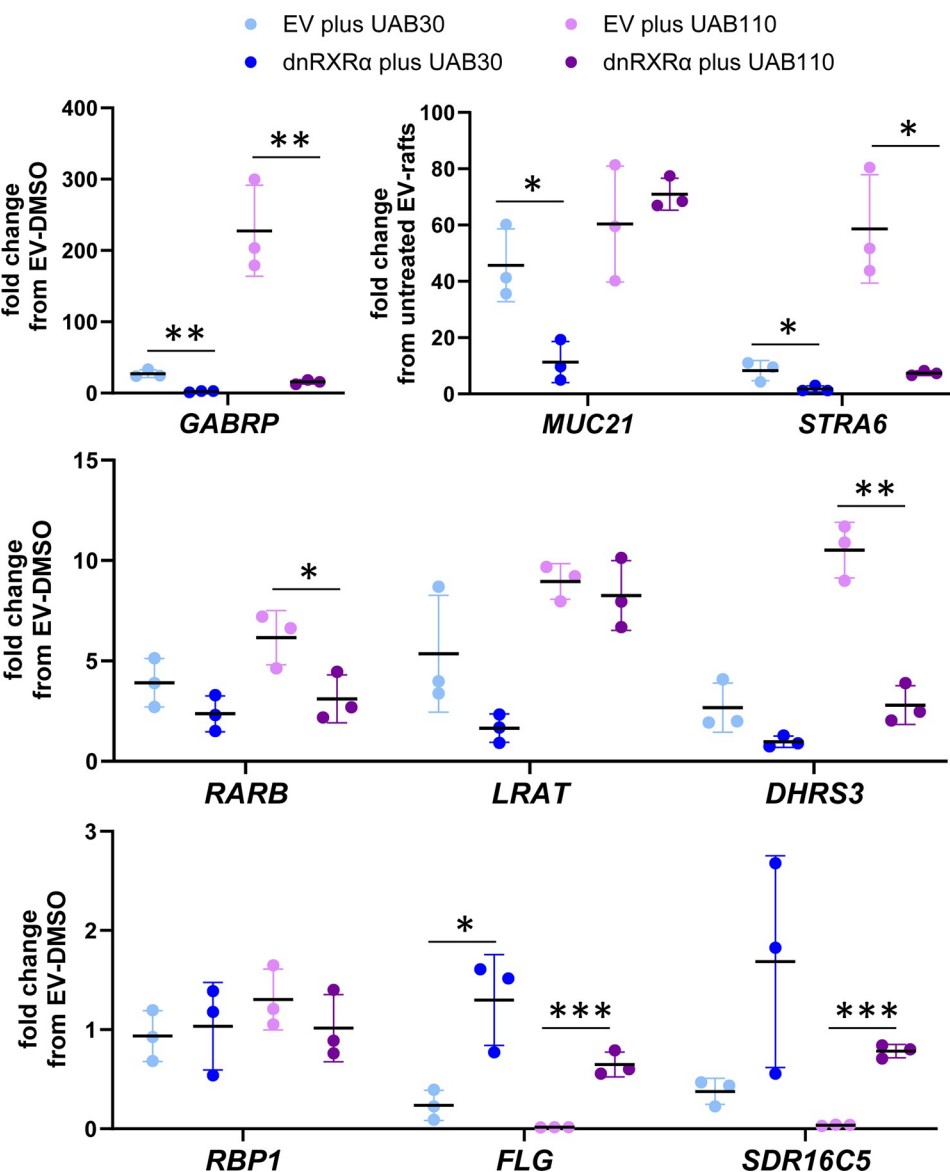

**Fig 3. Impact of dnRXRα on gene expression in skin rafts treated with UAB rexinoids.** QPCR analysis was performed on RNA isolated from three individual rafts. The effects of UAB30 (2 μM) and UAB110 (0.2 μM) were compared in EV-rafts *versus* dnRXRα-rafts. *Error bars* represent mean ± STD. The two-tailed *p* value was determined using GraphPad Prizm 8. Statistically significant changes are indicated by *$p<0.05$; **$p<0.01$; ***$p<0.001$. *GABRP*, γ-aminobutyric acid (GABA) receptor subunit π; *MUC21*, mucin 21; *STRA6*, stimulated by retinoic acid gene 6; *RARB*, retinoic acid receptor β; *LRAT*, lecithin retinol acyltransferase; *DHRS3*, dehydrogenase/reductase member 3 (also known as retSDR1); *RBP1*, cellular retinol binding protein type 1; *FLG*, filaggrin; *SDR16C5*, short-chain dehydrogenase/reductase 16C5.

of *MUC21* increased only by 8-fold, whereas the increase in other transcripts did not exceed 2-fold. The negatively regulated *FLG* and *SDR16C5* were significantly less downregulated in rexinoid-treated dnRXRα-rafts compared to EV-rafts.

As expected, UAB110 had a much stronger impact on expression of all genes in EV-rafts, with *GABRP* upregulated by 190-fold, *STRA6*—by 60-fold, and *MUC21*—by 40-fold. *RARB*, *LRAT* and *DHRS3* were upregulated by 6–10 –fold. Importantly, as was the case with UAB30, in the rafts overexpressing dnRXRα, the effects of UAB110 were greatly reduced, with *GABRP*

transcript showing only a 14-fold increase and the changes in other genes not exceeding 9-fold. *FLG* expression remained comparable to that in DMSO-treated rafts, and this was in agreement with the lack of changes in immunocytochemical staining of the differentiated granular layer (Fig 2, dnRXRα+UAB30, dnRXRα+UAB110).

Notably, not all of the genes tested were equally affected by the overexpression of dnRXRα. In skin rafts treated with UAB110, dnRXRα was ineffective in blocking the increase in transcript levels of *MUC21* and *LRAT*. Potentially, this could be due to changes in stratification pattern that led to enrichment of specific cell population expressing *MUC21* and *LRAT*. This appears to be true at least in the case of *MUC21*. The sponge-like appearance of skin rafts with large gaps between the cells was consistent with the upregulation of *MUC21*, which encodes a large cell surface glycoprotein that inhibits cell-cell and cell-matrix adhesion [31].

**Analysis of retinoid levels in dnRXRα–transduced rafts in the presence and absence of rexinoids.** To determine whether the lack of rexinoid-enhanced ATRA signaling in dnRXRα-rafts was linked to ATRA levels, we measured ATRA concentration in skin rafts. The basal levels of ATRA in DMSO-treated EV-rafts were $3.35 \pm 0.62$ pmol/g tissue (Fig 4A). Treatment of EV-rafts with 2 μM UAB30 increased ATRA concentration to $6.32 \pm 1.60$ pmol/g tissue, whereas 0.2 μM UAB110 raised ATRA to $15.22 \pm 1.92$ pmol/g of tissue. These results were consistent with our previously reported effects of UAB rexinoids on ATRA levels in skin rafts [15].

In comparison, the basal levels of ATRA in DMSO-treated dnRXRα-rafts were $2.28 \pm 0.29$ pmol/g tissue, and rexinoid treatment resulted in little to no change in ATRA concentration, with $2.41 \pm 0.41$ pmol/g tissue for UAB30-treated dnRXRα-rafts and $2.53 \pm 0.40$ pmol/g tissue for UAB110-treated dnRXRα-rafts. Thus, overexpression of dnRXRα prevented the rexinoid-induced increase in ATRA concentration.

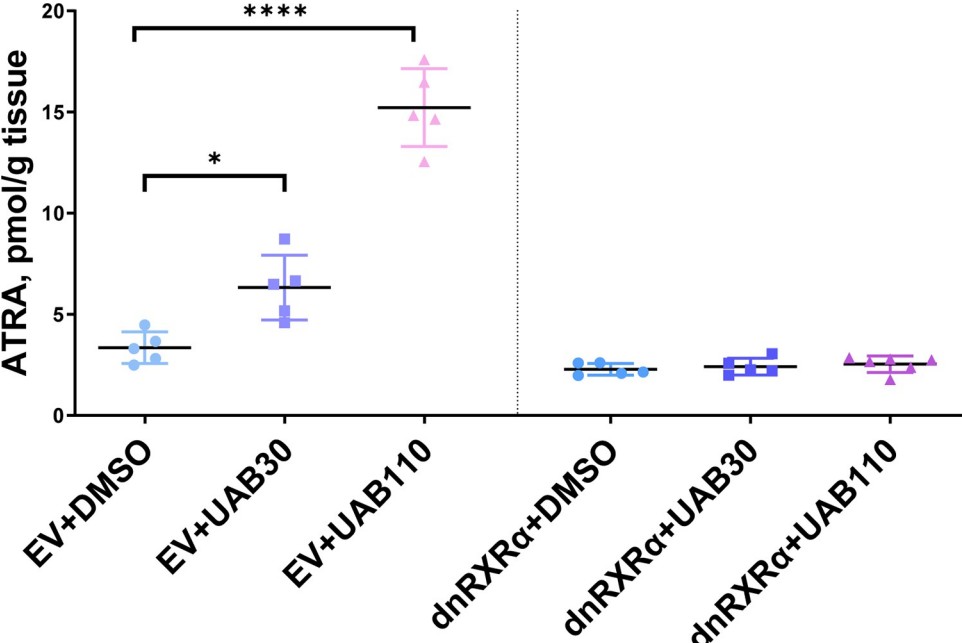

**Fig 4. Impact of dnRXRα on ATRA levels in skin rafts treated with UAB rexinoids.** EV-rafts and dnRXRα-rafts were treated with either vehicle (DMSO), UAB30 (2 μM) or UAB110 (0.2 μM). ATRA was analyzed by multistage tandem mass spectrometry, also known as multiple reaction monitoring cubed (MRM³), using an MRM³ transition of $m/z \ 301.1 \rightarrow m/z \ 205.1 \rightarrow m/z \ 159$ [32]; *$p<0.05$; ****$p<0.0001$.

**Global transcriptome analysis of dnRXRα–transduced rafts in the presence and absence of rexinoids.** To evaluate the global impact of dnRXRα overexpression on rexinoid signaling in skin rafts, we performed RNA-seq analysis. First, we examined the effects of rexinoids on gene expression patterns in EV-rafts. The light green oval in Fig 5A and the dark green oval in Fig 5B summarize the differentially expressed genes (DEG) (false discovery rate less than 0.05) in EV-rafts treated with 2 μM UAB30 compared to vehicle (DMSO)-treated EV-rafts. The number of upregulated DEGs in light green oval was 717 (562+10+3+16+1+125), and the number of downregulated DEGs—388 (318+10+24+6+30). The complete lists of DEGs can be found in S1–S5 Tables, with those in the overlapping areas of Venn diagrams–in S6–S9 Tables.

Fig 6A and 6B show that a ten-fold lower concentration of UAB110 (0.2 μM) resulted in 1680 (1306+7+2+19+346) upregulated DEGs and 1162 (824+9+68+4+23+234) downregulated DEGs, in agreement with the previously reported higher potency of UAB110 [15]. A subset of 905 DEGs was shared between UAB30 and UAB110-treated samples (S1 Table), with the rest of the DEGs being unique for each rexinoid.

We then asked how many of these DEGs were targets of RXRα signaling. Based on the analysis of ATRA-specific genes, we reasoned that the amplitude of transcriptional response of such genes will be diminished by overexpression of dnRXRα. The light blue ovals in Figs 5 and 6 summarize DEGs that were still upregulated by UAB30 (768) and UAB110 (617) despite the overexpression of dnRXRα (S2 Table). The dark blue ovals show how many DEGs were still downregulated: 307 for UAB30 and 385 for UAB110. This suggested that there were off-target effects of UAB rexinoids or that the regulation of these genes was not dependent on the AF-2 domain of RXR.

To address this question, we examined the effect of dnRXRα overexpression on the transcriptional profile of skin rafts treated with vehicle only. Gray ovals in Figs 5 and 6 summarize DEGs that were upregulated (100) in dnRXRα-rafts, and red ovals summarize DEGs that were downregulated (213), (S3 Table). Interestingly, among the DEGs in vehicle-treated dnRXRα-rafts were well-established targets of ATRA signaling including *COL1A1* [33], *CXCL14* [34], *KRT2* [35], *KLK13* [36], *CCN1* [37], *SCD* [38], *S100P* [39], *EGR1* [40], etc. Thus, signaling potency of the resident endogenous ATRA in keratinocytes was diminished by overexpression of dnRXRα. Together, these results suggested that overexpression of dnRXRα altered the

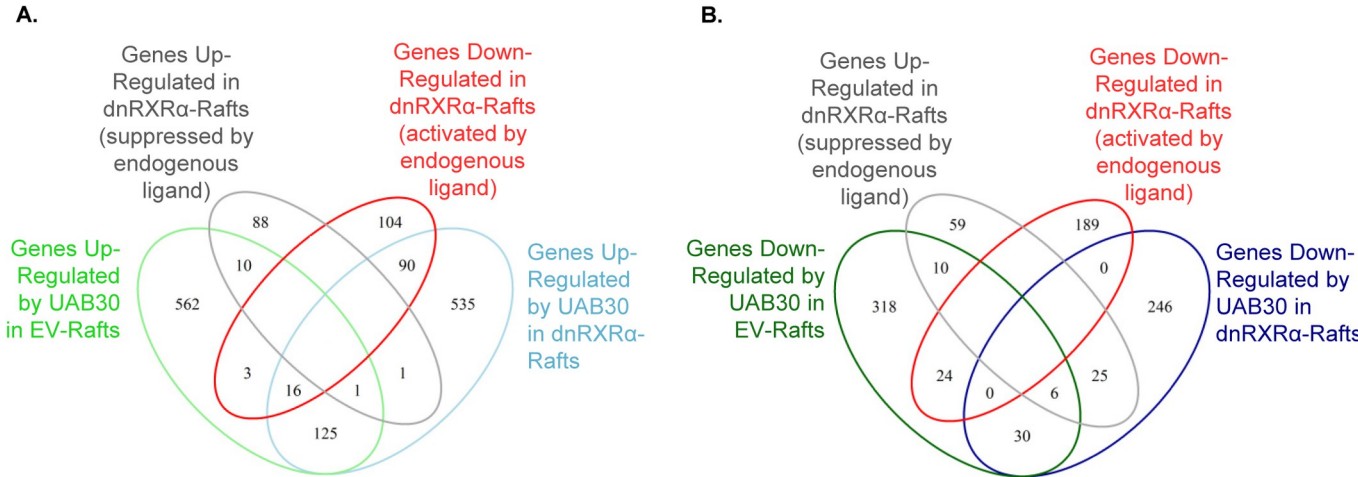

**Fig 5.** Venn diagrams of genes upregulated (A, S1 Table) and downregulated (B, S2 Table) by UAB30 in EV-rafts (green ovals) and dnRXRα-rafts (blue ovals). Genes in the overlap of green and blue areas retain responsiveness to UAB30 treatment despite the overexpression of dnRXRα. Genes in the overlaps of green and blue ovals with red (activated by endogenous ligand) or gray (suppressed by endogenous ligand) ovals constitute targets common between UAB30 and endogenous RXRα ligand.

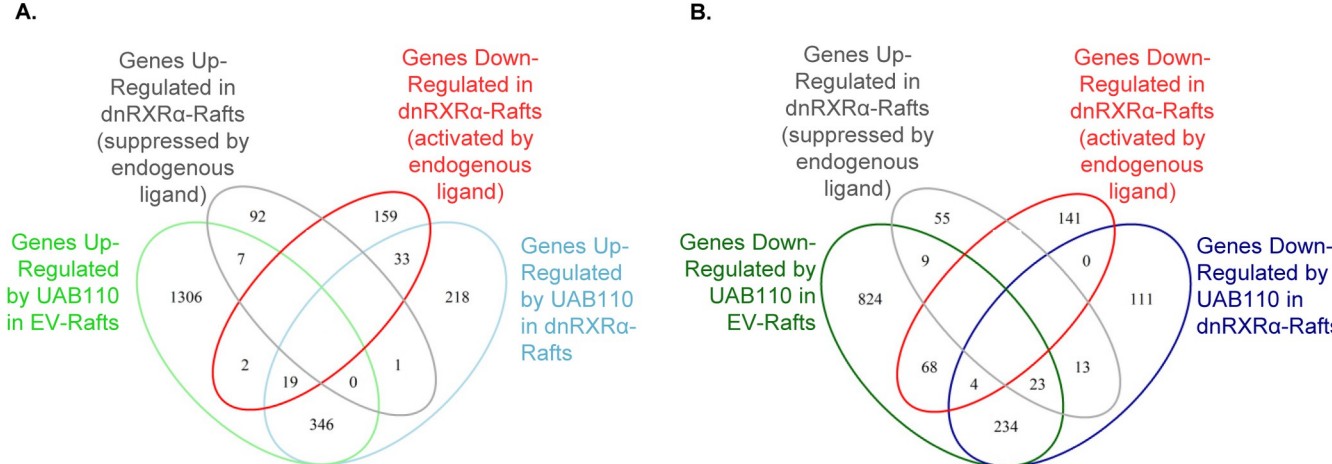

**Fig 6.** Venn diagrams of genes upregulated (A, S4 Table) and downregulated (B, S5 Table) by UAB110 in EV-rafts (green ovals) and dnRXRα-rafts (blue ovals). Genes in the overlap of green and blue areas retain responsiveness to UAB30 treatment despite the overexpression of dnRXRα. Genes in the overlaps of green and blue ovals with red (activated by endogenous ligand) or gray (suppressed by endogenous ligand) ovals constitute targets common between UAB110 and endogenous RXRα ligand.

baseline expression levels of a specific subset of genes, which have not displayed statistically significant changes in rexinoid-treated EV-rafts, but have fallen into the statistically significant group upon treatment with rexinoids in dnRXRα-rafts relative to their baseline levels in vehicle-treated dnRXRα-rafts.

This explains the unexpectedly small overlap between those genes that qualified as DEGs in response to UAB rexinoid treatment in EV-rafts and dnRXRα-rafts. Specifically, for UAB30 treatment, there were 1105 DEGs in EV-rafts compared with 1075 DEGs in dnRXRα-rafts, but the overlap constituted only 213 DEGs (S4 Table). For UAB110, there were 2842 DEGs in EV-rafts compared to 1002 DEGs in dnRXRα-rafts, with the overlap of only 653 DEGs (S5 Table). Some DEGs changed their directions in EV-rafts and dnRXRα-rafts. Thirty five genes were up in UAB30-treated EV-rafts but down in dnRXRα-rafts or vice versa and 27 DEGs behaved the same way in UAB110-treated rafts.

To identify potential off-target effects of UAB rexinoids, we plotted the expression levels of DEGs as a ratio of their levels in rexinoid-treated dnRXRα-rafts to rexinoid-treated EV-rafts, with a ratio of 1 indicating no change upon overexpression of dnRXRα (Fig 7). This allowed us to view the extent by which rexinoid effects were diminished by dnRXRα for each DEG. For the majority of these transcripts the ratio was below 1 (Fig 7, S10 Table), indicating that the deletion of AF-2 domain reduced the effects of synthetic rexinoids. To examine which genes were the least affected by overexpression of dnRXRα and could represent potential off-target effects of rexinoids, we set a cutoff of gene expression ratio at 0.8 (Fig 7), selecting a group of DEGs which had a decrease in response by 20% or less. For UAB30, this group consisted of 42 genes, and for UAB110–144 genes. There were only 13 DEGs that overlapped between UAB30 and UAB110 upregulated genes (*AKR1C2*, *C4orf19*, *MME*, *PAPPA*, *RPRML*, *S100A12*, *S100P*, *SDCBP2*, *SEMA5A*, *SLPI*, *TMPRSS11A*, and *ZNF812P*) and none that overlapped between the downregulated genes. For these 13 DEGs, the ratio of their expression levels in rexinoid-treated dnRXRα-rafts relative to rexinoid-treated EV-rafts was close to 1, indicating that their transcription was essentially unaffected by overexpression of dnRXRα. However, this group included established targets of ATRA signaling *S100P* [39], *PAPPA* [41], *MME* [42], and *SLPI* [43] that were also sensitive to the endogenous ligand as discussed above.

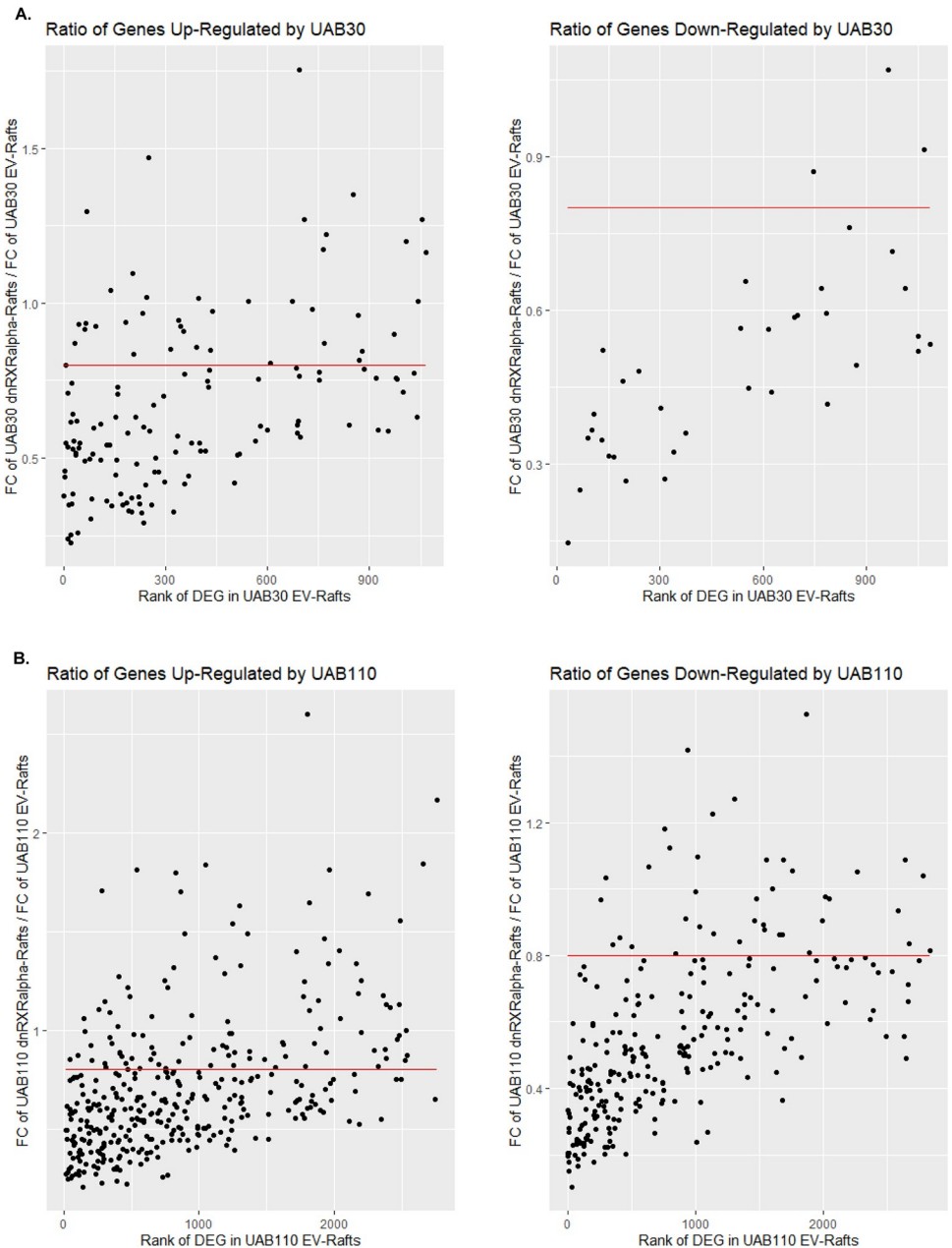

**Fig 7.** Ratio of DEGs in dnRXRα- and EV-rafts with and without UAB30 (A) and UAB110 (B) treatment. For rafts treated with UAB30 (A) or UAB110 (B) the fold change (FC) of DEGs in dnRXRα-rafts were divided by the FC of EV-rafts to create a ratio of expression. A ratio of 1 indicates no change in FC upon overexpression of dnRXRα. A ratio closer to 0 indicates a greater FC in EV-rafts, while a ratio greater than 1 indicates a greater FC in dnRXRα-rafts. A cutoff of 0.8 (red line) for the gene expression ratio was used to determine potential off-target genes for UAB30 and UAB110.

This suggested that some of the genes with a ratio of 1 were bona fide targets of ATRA, and that the remaining levels of the wild-type endogenous RXRα-RAR heterodimers and/or relatively low levels of resident keratinocyte ATRA were sufficient to maintain their expression.

The remaining genes in the group of 13 that were not affected by dnRXRα overexpression included *AKR1C2, C4orf19, RPRML, SDCBP2, TMPRSS11A,* and *ZNF812P*. We have not

found any evidence of ATRA-mediated regulation of these genes in the literature. Upregulation of these genes in rexinoid-treated rafts might be considered an "off-target" effect of rexinoids. Alternatively, the effects of ATRA on their expression have not yet been investigated.

Overall, a total of 17 genes upregulated by UAB30 and 51 genes upregulated by UAB110 were unaffected by overexpression of dnRXRα (a ratio of dnRXRα-rafts/EV-rafts expression levels equal 1 or above) (S10 Table). As discussed above, some of these genes are known targets of ATRA; therefore, the number of potential off-target genes is smaller than it appears.

## Discussion

This work is a continuation of our studies aiming to elucidate the mechanism of action of UAB series of synthetic rexinoids and the process of RXR regulation by differential ligand response. One of the questions remaining in the field is how specific rexinoid actions are and to what degree RXR protein is required for that. As we have shown previously, in organotypic skin epidermis, UAB30 and UAB110 rexinoids preferentially activate the RXR-RAR signaling pathway, resulting in increased levels of ATRA [14, 15]. Here, we investigated whether RXRα is necessary for mediating the increase in ATRA concentration and signaling. We chose to address this question by overexpressing dnRXRα mutant that can still dimerize with RAR and bind the rexinoids but is unable to communicate the activating binding of rexinoids to RAR due to the lack of the transactivating AF-2 domain. Skin rafts overexpressing dnRXRα have residual amounts of native RXRα, enabling keratinocytes to proliferate and differentiate into the stratified epidermis. In fact, overexpression of dnRXRα resulted in minimal changes to skin rafts' histology consistent with a mild decrease in ATRA signaling as indicated by the small changes in expression levels of ATRA target genes as well as the mild decrease in ATRA concentration in dnRXRα-rafts compared to EV-rafts.

The effects of UAB30 and UAB110 on histology of EV-raft were consistent with our previously published studies [14, 15], with increased proliferation of basal cells and reduced thickness of granular and cornified layers. In contrast, these effects of UAB30 and UAB110 were significantly diminished in dnRXRα-rafts. Interestingly, rexinoid-treated dnRXRα-rafts still had somewhat greater basal cell proliferation compared to DMSO-treated rafts, suggesting that dnRXRα was more effective in augmenting RXR signaling in the stratified layers than in the basal cell layer. The reason for this difference is unclear, but it could be due, in part, to the differential expression pattern of retinoid receptors in the epidermis. While RXRα is known to be evenly expressed throughout the epidermis, RARγ is more abundant in suprabasal layers, whereas RARα is enriched in basal layer [44].

Importantly, the upregulation and downregulation of the known ATRA target genes (*GABRP*, *STRA6*, *RARB*, *DHRS3*, *FLG*, *SDR16C5*) by rexinoids was nearly abolished in dnRXRα-rafts. Surprisingly, the upregulation of two other ATRA target genes, *LRAT* and *MUC21*, by UAB110 was not affected by dnRXRα overexpression. As mentioned in the *Results*, the lack of changes in *MUC21* levels could be potentially explained by the increased number of *MUC21*-expressing cells, whereas in the case of *LRAT*, the remaining levels of native RXRα could be sufficient for the more transcriptionally potent UAB110. Alternatively, these two ATRA target genes may be more predominantly regulated by RXR-RAR heterodimer formation upon ATRA binding to RAR and not the AF-2 function.

Critical evidence in support of the role of RXRα in mediating the increase in ATRA signaling came from the measurements of ATRA concentration in rexinoid-treated dnRXRα-rafts *versus* EV-rafts. Consistent with our previous report [15], ATRA concentration in EV-rafts rose nearly 2-fold and 4.5-fold upon treatment with UAB30 and UAB110, respectively; however, no significant increase in ATRA concentration was detected in rexinoid-treated dnRXRα-rafts. Thus, the

AF-2 domain of RXRα was necessary for mediating the rexinoid-induced rise in ATRA concentration and signaling. This implies that the rexinoid-induced rise in ATRA concentration is highly dependent on coactivator binding to the RXR AF-2 site that is formed by Helices 3, 4 and 12. In our previous report we performed biophysical analysis of RXRα ligand binding domain in complex with UAB30 and UAB110 [15]. UAB110 had higher affinity and reduced the overall dynamics of RXR to a greater extent when compared to UAB30. This included when a GRIP-1 coactivator peptide was bound [15]. Without the RXR coactivator-binding site, there are no differential levels of transactivation for increased ATRA concentration.

The comparative analysis of the global transcriptomes of vehicle (DMSO)-treated EV-rafts and dnRXRα-rafts produced an unanticipated finding: overexpression of dnRXRα resulted in reduced levels of ATRA-regulated transcripts. Thus, the signaling potency of resident ATRA was diminished. Previous in vitro studies showed that the deletion of AF-2 domain in RXR induces its ability to interact with the transcriptional corepressors NCoR and SMRT [45, 46]. Thus, the effects observed in the presence of this RXR mutant could be, in part, due to induced repression rather than loss of activation. However, if indeed the dnRXRα/RAR complex does promote some corepressor binding, the effects of this binding are very mild since it has very little effect on the proliferation and differentiation of dnRXRα-overexpressing skin raft cultures. Indeed, ATRA levels in dnRXRα+DMSO skin rafts are very close to those in empty vector-transfected cells +DMSO. If there was a significant decrease in ATRA levels, the keratinocytes would not proliferate and form stratified layers. Also, out of nine ATRA-regulated genes in Fig 1C only one, *MUC21*, is downregulated in dnRXRα-rafts, again indicating a very low level of repression. Furthermore, the upregulation of other ATRA-regulated genes by rexinoids is diminished in the presence of dnRXRalpha, but there is still a slight gradient effect when going from the moderate agonist UAB30 to the strong agonist UAB110 for genes such as *FLG* and *SDR16C5*. For other genes there is minimal difference between the two rexinoids (*GABRP*, *STRA6*, *RARB*, *DHRS3*) but the overall effect is not repression. Thus, our results seem to be more complicated than just corepressor enhanced binding to the dnRXRα-RAR heterodimer. In addition, it is currently unknown how the presence of RXR agonist would affect the binding of corepressors to dnRXRα.

A number of studies suggested the existence of endogenous RXR ligands. Docosahexaenoic acid (DHA) was proposed to act as endogenous RXR ligand in brain [47], but it was argued that the pool of unesterified DHA is too low under physiological conditions [48]. The long-chain fatty acid C24:5 was identified as the most likely endogenous ligand for RXRα in hematopoietic cells [49]. Phytanic acid was also suggested to bind RXR [50, 51], but evidence to support its physiological relevance is lackingA vitamin A metabolite, 9-*cis*-13,14-dihydroretinoic acid (9-*cis*-13,14-DHRA), was reported to serve as an endogenous RXR ligand in mouse serum, brain, and liver [48] and also in human serum [52]. If ATRA signaling through RXR-RAR heterodimers in skin raft cultures was potentiated by an endogenous RXR ligand, this would explain the slightly reduced potency of ATRA signaling in dnRXRα-rafts.

One of the primary goals was also to evaluate potential "off-target" effects of UAB synthetic rexinoids. Since our data suggested that UAB rexinoids act by enhancing ATRA signaling, we reasoned that the most likely "off-target" genes will be those that are not subject to ATRA regulation, and thus, overexpression of dnRXRα mutant will have little or no effect on their transcript levels. To our surprise, many of the genes that were unaffected by dnRXRα overexpression (ratio of expression levels in rexinoid-treated dnRXRα-rafts to rexinoid-treated EV-rafts above 0.8) turned out to be *bona fide* ATRA targets. In fact, this is in agreement with the known differential sensitivity of ATRA target genes to ATRA concentration [53, 54] and suggests that the residual levels of native RXRα-RAR heterodimers together with *in situ*

available ATRA are sufficient for transcriptional activation of these genes, and treatment with RXRα agonists, which raise ATRA levels, does not enhance their transcription.

Eleven (10+1) DEGs in the overlap of green and gray ovals in Fig 5A (*CCN1*, *EGR1*, *FOS*, *HES1*, *RHOB*, *EDN1*, *IGFL1*, *TAGLN*, *COMT*, *AKT1*, and *COL1A1*) (S3 and S6 Tables) were upregulated in dnRXRα-rafts but were induced by UAB30 in EV-rafts. Out of these 11 DEGs, one (*COL1A1*) was also located in the overlap with blue oval, which included DEGs upregulated by UAB30 in dnRXRα-rafts. *COL1A1* displayed an unusual behavior: it was upregulated in vehicle-treated dnRXRα-rafts, suggesting that this gene was naturally suppressed in keratinocytes, but UAB30 treatment overruled this suppression and enhanced *COL1A1* expression not only in EV-rafts containing normal levels of RXRα-RAR, but also in dnRXRα-rafts, which presumably had reduced amount of RXRα-RAR. Others have shown that *Col1a1* expression was increased 3-fold in vitamin A deficient rat embryos [33], consistent with it being suppressed by RXRα-RAR signaling.

Another singular DEG in the overlap of gray and blue ovals in Fig 5 was *CXCL14*. Its expression was also enhanced in dnRXRα-rafts, consistent with it being downregulated by ATRA as reported for adipocytes [34], but unlike *COL1A1*, upregulation of *CXCL14* by UAB30 reached significance only in dnRXRα-rafts, perhaps due to its altered baseline levels relative to EV-rafts.

An interesting group of genes was identified in the overlap of green, blue and red ovals in Fig 5A (S6 Table). These DEGs were upregulated by UAB30 in both EV-rafts (green) and dnRXRα-rafts (blue), and were downregulated in vehicle-treated EV-rafts (red) (S3C Table). This suggested that these 16 genes continued to respond to UAB30 even at reduced levels of wild-type RXRα-RAR heterodimers. The 16 genes included *MME*, *PAPPA*, *S100P*, *SLPI*, which are known as targets of ATRA signaling [39, 41–43]. The same four genes were also upregulated by UAB110 in EV-rafts as well as in dnRXRα-rafts. Alternatively, it is possible that the RXRα-dependent regulation of these genes is not dependent on the AF-2 function. Thermodynamics studies of rexinoid binding to RXRα-LBD have demonstrated the increased thermodynamic stability of RXR homodimers correlated with higher affinity ligands [15]. This same group has recently demonstrated the same for RXR-RAR heterodimers (unpublished data in communication with Z. Yang and M. Renfrow). It is possible that these genes are responding to an increase in ligand-induced RXR-RAR heterodimer stability or increased coactivator binding to the RAR AF-2 site. It could also be that agonist binding to native RXRα and dnRXRα promotes more RXRα-RAR heterodimer formation over RXR homodimer and even tetramer formation in vivo.

Overall, our data suggested that over 95% of the DEGs in rexinoid-treated rafts constitute direct or indirect RXRα-regulated genes. This number was derived by dividing the number of DEGs that had a decrease in response in dnRXRα-rafts by 20% or less over the total number of DEGs in EV-rafts. For UAB30, this ratio was 42/1105 (717 up+ 388 down) = 3.8% DEGs, and for UAB110–144/2842 (1680 up + 1162 down) = 5% DEGs. These DEGs included genes known to be regulated by ATRA (*MME*, *PAPPA*, *S100P*, *SLPI*), suggesting the actual off-target effects are potentially smaller or as stated above could potentially reflect non-AF-2-dependent mechanisms of gene regulation by RXR. The combined approaches we report here for the analysis of dnRXRα with respect to AF-2 has the potential to be further expanded to elucidate differential RXRα activation mechanisms in our skin raft cultures and other model systems. As ATRA levels are known to be reduced in certain epithelial pathologies, treatment with UAB30 and UAB110 may represent a promising therapy for normalizing the endogenous ATRA concentration and signaling with minimal side effects in epithelial tissues.

## Supporting information

**S1 Fig. RXR western blot.**
(JPG)

**S2 Fig. Actin western blot.**
(JPG)

**S3 Fig. Ponceau stained blot.**
(JPG)

**S1 Table. DEGs in UAB30- and UAB110-treated EV-rafts.**
(XLSX)

**S2 Table. DEGs in UAB30 and UAB110-treated dnRXRα rafts.**
(XLSX)

**S3 Table. 313 DEGs in dnRXRα-DMSO vs EV-DMSO rafts.**
(XLSX)

**S4 Table. DEGs in UAB30-treated EV- and dnRXRα-rafts.**
(XLSX)

**S5 Table. DEGs in UAB110-treated EV- and dnRXRα-rafts.**
(XLSX)

**S6 Table. List of DEGs for Venn diagram in Fig 5A (upregulated by UAB30).**
(XLSX)

**S7 Table. List of DEGs for Venn diagram in Fig 5B (downregulated by UAB30).**
(XLSX)

**S8 Table. List of DEGs for Venn diagram in Fig 6A (upregulated by UAB110).**
(XLSX)

**S9 Table. List of DEGs for Venn diagram in Fig 6B (downregulated by UAB110).**
(XLSX)

**S10 Table. UAB30-upregulated genes the least affected by the expression of dnRXRα.**
(XLSX)

## Acknowledgments

We are grateful to Dr. Kirill Popov, Ph. D., Professor in the Department of Biochemistry and Molecular Genetics at the University of Alabama at Birmingham Heersink School of Medicine for excellent suggestions on evaluating the off-target effects of rexinoids and helpful discussions.

## Author Contributions

**Conceptualization:** Olga V. Belyaeva, Venkatram R. Atigadda, Kelli R. Goggans, Matthew B. Renfrow, Laszlo Nagy, Natalia Y. Kedishvili.

**Data curation:** Olga V. Belyaeva, Alla V. Klyuyeva, Ansh Vyas, Wilhelm K. Berger, Laszlo Halasz, Jianshi Yu, Aja Slay, Kelli R. Goggans, Matthew B. Renfrow, Maureen A. Kane, Laszlo Nagy, Natalia Y. Kedishvili.

**Formal analysis:** Olga V. Belyaeva, Ansh Vyas, Wilhelm K. Berger, Laszlo Halasz, Jianshi Yu, Kelli R. Goggans, Matthew B. Renfrow, Maureen A. Kane, Laszlo Nagy, Natalia Y. Kedishvili.

**Funding acquisition:** Venkatram R. Atigadda, Matthew B. Renfrow, Maureen A. Kane, Laszlo Nagy, Natalia Y. Kedishvili.

**Investigation:** Olga V. Belyaeva, Alla V. Klyuyeva, Ansh Vyas, Wilhelm K. Berger, Laszlo Halasz, Jianshi Yu, Aja Slay, Kelli R. Goggans, Maureen A. Kane, Laszlo Nagy, Natalia Y. Kedishvili.

**Methodology:** Olga V. Belyaeva, Alla V. Klyuyeva, Ansh Vyas, Wilhelm K. Berger, Laszlo Halasz, Jianshi Yu, Aja Slay, Kelli R. Goggans, Maureen A. Kane, Laszlo Nagy.

**Project administration:** Maureen A. Kane, Laszlo Nagy, Natalia Y. Kedishvili.

**Resources:** Venkatram R. Atigadda, Maureen A. Kane, Laszlo Nagy, Natalia Y. Kedishvili.

**Supervision:** Maureen A. Kane, Laszlo Nagy, Natalia Y. Kedishvili.

**Validation:** Olga V. Belyaeva, Alla V. Klyuyeva, Wilhelm K. Berger, Laszlo Halasz, Jianshi Yu, Aja Slay, Kelli R. Goggans, Maureen A. Kane, Laszlo Nagy, Natalia Y. Kedishvili.

**Visualization:** Olga V. Belyaeva, Alla V. Klyuyeva, Wilhelm K. Berger, Laszlo Halasz, Jianshi Yu, Aja Slay, Kelli R. Goggans, Maureen A. Kane, Laszlo Nagy, Natalia Y. Kedishvili.

**Writing – original draft:** Olga V. Belyaeva, Alla V. Klyuyeva, Wilhelm K. Berger, Laszlo Halasz, Jianshi Yu, Kelli R. Goggans, Laszlo Nagy, Natalia Y. Kedishvili.

**Writing – review & editing:** Olga V. Belyaeva, Jianshi Yu, Venkatram R. Atigadda, Kelli R. Goggans, Matthew B. Renfrow, Maureen A. Kane, Laszlo Nagy, Natalia Y. Kedishvili.

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
