## [Decision Letter · Decision Letter 0]

8 Jan 2024

PONE-D-23-37860The Retinoid X Receptor has a critical role in synthetic rexinoid-induced increase in cellular All-Trans-Retinoic AcidPLOS ONE

Dear Dr. Kedishvili,

Thank you for submitting your manuscript to PLOS ONE. After careful consideration, we feel that it has merit but does not fully meet PLOS ONE’s publication criteria as it currently stands. Therefore, we invite you to submit a revised version of the manuscript that comprehensively addresses the points raised during the review process.

We look forward to receiving your revised manuscript.

Kind regards,

Michael Schubert

Academic Editor

PLOS ONE

Journal Requirements:

"V. R. A., M. B. R., L. N. and N. Y. K. were supported by the UAB O’Neil Comprehensive Cancer Center https://www.onealcanceruab.org and the National Cancer Institute https://www.cancer.gov grant CA210946. L. N. was also supported by the National Institute of Diabetes and Digestive and Kidney Diseases https://www.niddk.nih.gov grant DK115924; M. A. K. was supported by the National Institute of Arthritis and Musculoskeletal and Skin Diseases https://www.niams.nih.gov grant R01AR074846; University of Maryland, School of Pharmacy, P23474; and University of Maryland, School of Pharmacy Mass Spectrometry Center, SOP1841-IQB2014. N. Y. K. was also supported by the National Institute of Arthritis and Musculoskeletal and Skin Diseases https://www.niams.nih.gov grant R01AR076924."

4. Please expand the acronym “NIH NCI”, "NIH NIDDK" and "NIH NIAMS" (as indicated in your financial disclosure) so that it states the name of your funders in full.

5. We note that you have referenced  (unpublished data in communication with Z. Yang and M. Renfrow) which has currently not yet been accepted for publication. Please remove this from your References and amend this to state in the body of your manuscript: (unpublished data in communication with Z. Yang and M. Renfrow) as detailed online in our guide for authorshttp://journals.plos.org/plosone/s/submission-guidelines#loc-reference-style

Reviewers' comments:

Reviewer's Responses to Questions

**Comments to the Author**

1. Is the manuscript technically sound, and do the data support the conclusions?

Reviewer #1: Yes

Reviewer #2: Partly

2. Has the statistical analysis been performed appropriately and rigorously? 

Reviewer #1: Yes

Reviewer #2: Yes

3. Have the authors made all data underlying the findings in their manuscript fully available?

Reviewer #1: Yes

Reviewer #2: Yes

4. Is the manuscript presented in an intelligible fashion and written in standard English?

Reviewer #1: Yes

Reviewer #2: Yes

5. Review Comments to the Author

Reviewer #1: This study aims to elucidate the mechanism underlying the increased concentrations of all-trans retinoic acid (atRA), potentially induced by RXR and its ligands such as UAB30 and UAB110, by using a skin organoid culture model established by these investigators. Utilization of a dominant negative mutant of RXR alpha (dnRXR-alpha), which lacks the AF-2 domain essential for ligand-dependent gene regulation, demonstrated that this AF-2 domain is necessary for UAB-mediated increases in atRA concentrations. Global gene expression analysis showed altered expression of atRA-regulated genes, and supported their suggestion of potential endogenous RXR ligands. Data also revealed several possible off-target genes for each RXR ligand. Overall, this study provides evidence for the proposed mechanism of UAB RXR ligands. However, since the mechanism of gene expression involves multiple regulatory pathways and transcription factors that must act coordinately (dynamic and kinetic) in a physiological context, it may be helpful to discuss the potential complication, such as the timing (duration of treatment) and condition of the experiments, that may also contribute to changes in the expression of these “RA-regulated” genes. Further, some experimental conditions and technical issues need to be clarified (see below).

1) Organotypic skin raft culture:

i) Are donor foreskins from a single individual used to generate cultures for each replicate of an experiment, or was pooled donor tissue used to generate cultures? This information would be helpful to the understanding of possible biological/individual variability that this model is subject to.

ii) Is the skin raft model maintained in media that have certain critical reagents (e.g. fetal bovine serum) known to contain retinoids and other hormones? If so, the authors may need to carefully consider the potential contributions and confounding effects of certain components in the serum.

2) RNA seq studies: What was considered a biological replicate and how many biological replicates were performed?

3) Given the altered RA concentrations, it would be useful to discuss whether the expression of classically known RA synthesis and/or catabolism genes was affected significantly?

Reviewer #2: Tracking #: PONE-D-23-37860 – Belyaeva et al.

The manuscript entitled “The Retinoid X Receptor has a critical role in synthetic rexinoid-induced increase in cellular All-Trans-Retinoic Acid” attempts to address interesting new issues regarding the action mode of synthetic rexinoids in organotypic human epidermis. This study continues research published by the authors in the past.

The title clearly announces the paper's main conclusion and the introduction allows for a good understanding of the subject.

Although this study demonstrates that rexinoids mediate increases in retinoic acid levels via RXR in human epidermis, it suffers from conceptual flaws which means that some of the conclusions and interpretations formulated by the authors should not be done.

Two major points are worth noting. The first concerns the choice of using an RXR mutant deleted from helix H12 (dnRXR). Although the authors' choice of a receptor that loses the ability to be activated by a rexinoid agonist seems relevant for assessing RXR's contribution to rexinoid-mediated effects, H12 deletion proves too drastic for the purposes of this study. This mutant was insufficiently controlled by the authors. This makes for a paradoxical work with very uneven conclusions and interpretations of values. Indeed, it has been shown that such a deletion induces the ability of RXR to interact with the transcriptional corepressors NCoR and SMRT; wild-type RXR does not interact with these corepressors (Hu and Lazar, 1999, Nature 402:93-6). This results in dnRXR-mediated transcriptional repression, even in the context of heterodimers containing dnRXR (and thus the one formed with RAR). So the effects observed in the presence of this RXR mutant are probably due to induced repression rather than loss of activation. The interpretation of the results obtained needs to be re-evaluated. Not only does the assertion that an endogenous rexinoid ligand exists and is active (see pages 11 and 14) need to be qualified, but also the transcriptomics analyses relating to dnRXR need to be reconsidered. This point seems to me all the more important as the authors, in the perspectives at the end of the discussion, stress the potential of such a mutant for the continuation of their work. Another point concerns the presence of endogenous wild-type RXR, which competes with exogenous dnRXR. Authors sometimes mention this, but more often than not ignore it. The presence of wild-type RXR should be considered in the interpretation of all data. For this reason, the text should also be more precise. Moreover, these imprecisions make the discussion section confusing and speculative.

Overall, the authors' main conclusion regarding the involvement of RXR in the rexinoid-induced increase in retinoic acid levels in human epidermis is valid, but the whole text needs to be rewritten taking into account the previous indications to gain in precision and clarity. Based on the experiences presented, this article will be of certain interest after further writing.

The following is a list of points that should be clarified and/or improved:

1/ The choice of the H12-deleted RXR mutant is debatable. Other options may be considered. Hu and Lazar's paper shows that certain mutations in the hydrophobic groove of RXR (in H3 and H4, the domain mentioned by the authors on page 13) inhibit activation by preventing the recruitment of coactivators, while not allowing the binding of corepressors. Another possibility would be to use mutants of the RXR ligand-binding pocket that prevent rexinoid binding (Le Maire et al., 2022, JME, 69:377-390).

Also, if it is possible to use siRNAs in the model exploited in this study, siRNAs directed against RXRalpha, but also RARgamma, would address the questions posed by the authors.

2/ Gene expression analyses by qPCR (Fig 1 and 3) use only 3 replicates. This is perhaps insufficient for clear conclusions. For example for the effect of dnRXR on the expression of MUC21 (Fig 1C) a large variation is observed for the control condition which leads the authors to consider a significant difference between the control and the dnRXR condition, and this only for MUC21. This test should be repeated to confirm the interpretation made. In a more nuanced way, it seems to me that the expression of dnRXR causes a slight decrease in the expression of several of the genes studied.

3/ Although the reported results indicate an effect of dnRXR, and thus that its expression is effective, it is difficult to evaluate the relative expressions of endogenous wild-type and exogenous mutated forms of RXRalpha. The use of a tagged dnRXR would make it possible to differentiate the two proteins. Also the Western blot conditions could be optimized to improve the separation even if the molecular mass difference is small.

4/ For transcriptomics analyses, authors often refer to retinoid target genes. Are there data in this human epidermis model for retinoic acid? Such data would be valuable for being able to precisely correlate the effects of rexinoids to genes and pathways regulated by retinoic acid.

5/ It would be nice to have the structure of the UAB30 and UAB110 rexinoids, as well as their corresponding chemical formula.

6/ Typo: page 13 line 7, “overexpression of dnRXRa” instead of “RXRa”.

6. PLOS authors have the option to publish the peer review history of their article (what does this mean?). If published, this will include your full peer review and any attached files.

Reviewer #1: No

Reviewer #2: No

---

## [Author Response · Author response to Decision Letter 0]

28 Feb 2024

see attached file Response to reviewers

---

## [Decision Letter · Decision Letter 1]

17 Mar 2024

The Retinoid X Receptor has a critical role in synthetic rexinoid-induced increase in cellular All-Trans-Retinoic Acid

PONE-D-23-37860R1

Dear Dr. Kedishvili,

We’re pleased to inform you that your manuscript has been judged scientifically suitable for publication and will be formally accepted for publication once it meets all outstanding technical requirements.

Kind regards,

Michael Schubert

Academic Editor

PLOS ONE

Reviewer's Responses to Questions

**Comments to the Author**

1. If the authors have adequately addressed your comments raised in a previous round of review and you feel that this manuscript is now acceptable for publication, you may indicate that here to bypass the “Comments to the Author” section, enter your conflict of interest statement in the “Confidential to Editor” section, and submit your "Accept" recommendation.

Reviewer #1: All comments have been addressed

Reviewer #2: All comments have been addressed

2. Is the manuscript technically sound, and do the data support the conclusions?

Reviewer #1: Yes

Reviewer #2: Yes

3. Has the statistical analysis been performed appropriately and rigorously? 

Reviewer #1: Yes

Reviewer #2: Yes

4. Have the authors made all data underlying the findings in their manuscript fully available?

Reviewer #1: Yes

Reviewer #2: Yes

5. Is the manuscript presented in an intelligible fashion and written in standard English?

Reviewer #1: Yes

Reviewer #2: Yes

6. Review Comments to the Author

Reviewer #1: (No Response)

Reviewer #2: I would like to thank the authors for considering my earlier comments and suggestions in this revision.

In this new version of the manuscript, it is appreciated that the hypothesis of the action of an endogenous rexinoid has been abandoned. With regard to the choice of using an RXR mutant deleted from helix H12, the authors draw on bibliographical data to justify it. In particular, they point to the contribution of RXR in the interaction of the RXR-RAR heterodimer with transcriptional corepressors. Although a contribution has been reported using biophysical methods, the situation in the cellular context is quite different, and the relative interaction affinities of RXR and RAR with corepressors are very different, and the type of corepressor involved (NCoR or SMRT) may also play a role. In the context of this paper, debate on this point would be excessive. But I still have my doubts about the choice of this RXR mutant, which does not call into question the main conclusions reached.

7. PLOS authors have the option to publish the peer review history of their article (what does this mean?). If published, this will include your full peer review and any attached files.

Reviewer #1: No

Reviewer #2: No

---

## [Editor Report · Acceptance letter]

22 Mar 2024

PONE-D-23-37860R1 

PLOS ONE

Dear Dr. Kedishvili, 

I'm pleased to inform you that your manuscript has been deemed suitable for publication in PLOS ONE. Congratulations! Your manuscript is now being handed over to our production team.

Kind regards, 

on behalf of

Dr. Michael Schubert 

Academic Editor

PLOS ONE